# Impact on Physical Fitness of the Chinese CHAMPS: A Clustered Randomized Controlled Trial

**DOI:** 10.3390/ijerph16224412

**Published:** 2019-11-11

**Authors:** Zhixiong Zhou, Shiyu Li, Jun Yin, Quan Fu, Hong Ren, Tao Jin, Jiahua Zhu, Jeffrey Howard, Tianwen Lan, Zenong Yin

**Affiliations:** 1Institute for Sport Performance and Health Promotion, Capital University of Sports and Physical Education, Beijing 100191, China; yinjun@cupes.edu.cn (J.Y.); fuquan@cupes.edu.cn (Q.F.); lan13865352845@163.com (T.L.); 2Department of Kinesiology, Health and Nutrition, the University of Texas at San Antonio, San Antonio, TX 78249, USA; Shiyu.li@utsa.edu (S.L.); Jeffrey.Howard@utsa.edu (J.H.); Zenong.Yin@utsa.edu (Z.Y.); 3School of Sport Sciences, Beijing Sport University, Beijing 10089, China; renhong@bsu.edu.cn; 4College of Physical Education, Anhui Normal University, Wuhu 241000, China; asdjt@ahnu.edu.cn; 5College of Sports Science, Hefei Normal University, Hefei 230061, China; zhujiahua@hfnu.edu.cn

**Keywords:** cardiorespiratory fitness, socioecological model, physical functional training, high-intensity physical activity

## Abstract

Background: School physical activity (PA) policy, physical education curriculum, teacher training, knowledge of physical fitness, and parental support are among the key issues underlying the declining trend of physical fitness in children and adolescents. The Chinese CHAMPS was a multi-faceted intervention program to maximize the opportunities for moderate and vigorous physical activity (MVPA), and increase physical fitness in middle school students. The purpose of the study was to test whether the levels of modification in school physical education policy and curriculum incrementally influenced the changes in cardiorespiratory fitness and other physical fitness outcomes. Methods: This 8-month study was a clustered randomized controlled trial using a 2 × 2 factorial design. The participants were 680 7th grade students (mean age = 12.66 years) enrolled in 12 middle schools that were randomly assigned to one of four treatment conditions: school physical education intervention (SPE), afterschool program intervention (ASP), SPE+ASP, and control. Targeted behaviors of the Chinese CHAMPS were the student’s sedentary behavior and MVPA. The study outcomes were assessed by a test battery of physical fitness at the baseline and posttest. Sedentary behavior and MVPA were measured in randomly selected students using observations and accelerometry. Results: The terms contrasting the pooled effect of SPE, ASP, and SPE+ASP vs. Control, the pooled effect of SPE and SPE+ASP vs. ASP only, and the effect of SPE+ASP vs. ASP on CRF and other physical fitness outcomes were all significant after adjusting for covariates, supporting the study hypothesis. Process evaluation demonstrated high fidelity of the intervention in the targeted students’ behaviors. Conclusions: Chinese CHAMPS demonstrated the impact of varying the amount of MVPA and vigorous physical activity (VPA) on the physical fitness in middle school students in support of the need to increase the opportunity for PA in schools and to introduce high-intensity exercises in school-based PA programs. Modification of school policy, quality of physical education curriculum, and teacher training were important moderators of the improvement in physical fitness. (Trial registration: ChiCTR-IOR-14005388, the Childhood Health; Activity and Motor Performance Study).

## 1. Introduction

Over the past two decades, there has been a growing consensus that physical fitness is an important marker of cardiometabolic and skeletal health, cognitive development, and quality of life in children and adequate levels of physical activity (PA) being essential to develop and maintain a healthy level of physical fitness [1,2,3]. Although physical fitness and cardiorespiratory fitness (CRF) are sometimes used interchangeably, the former refers to a multitude of physiological attributes that characterize the human organization’s capabilities to perform at a satisfactory level in response to a physiologically demanding stress [2,4]. The attributes of physical fitness generally include endurance (CRF), speed, muscle strength, agility, flexibility, body height, and body composition. One study showed that CRF explained 3–15% variance in other attributes of physical fitness [5]. Many criterion-referenced standards have been established to assess the status of youth physical fitness in field-settings [6], such as the FitnessGram^®^ in the United States, the National Physical Fitness Test in China, and the EuroFit by the European Union [7]. Contemporary literature on the relationship between health and physical fitness has used CRF as a proxy of physical fitness in youth studies. CRF is consistently linked with improvement in cardiometabolic health, mental health and cognitive functions in children and adolescents, while the evidence on the relations with other attributes of physical fitness are still accumulating [1,8]. Therefore, there is a need to demonstrate PA intervention can impact CRF as well as other attributes of physical fitness in children.

Current PA guidelines recommend children and adolescents engage in a minimum of 60 min daily of moderate and vigorous physical activity (MVPA) which should include a variety of age-appropriate activities, and vigorous physical activity (VPA) 3 days a week [9]. Recent data show that 29.9% of Chinese school-age children (36.5% in grades 4–6, 32.5% in grades 7–9, and 27.7% in grades 10–12) met the PA recommendations [10]. Less than 20% of children in a national representative sample of American youth aged 12–17 [11] as well as a sample of adolescents aged 13–15 from 105 countries [12] met the PA recommendation. Marrow and colleagues found that American children who did not meet the PA recommendation were at increased risk of not meeting healthy physical fitness standards [13]. Finally, globally youth only spent a small portion of their time engaging in high-intensity PA [14].

There is a clear trend of the decline in PA and fitness and the increase in overweight and obesity in Chinese children over the past three decades [15,16,17]. The 2018 China Report Card on PA for children gave a grade of F, D+, D, and D+ to the overall PA levels (at least 60 min of MVPA daily in past week), active play (participation in unstructured/unorganized PA in last week), physical fitness (meeting excellent/good standard in the national fitness test), and supportive school PA policy and environment, respectively [18]. Finally, Chinese children ranked in the 60th percentile in the 20-m shuttle run (20MSR), a measure of CRF in children aged 9–17 out of 50 countries [19]. The epidemic of physical inactivity and declined physical fitness in Chinese children constitutes a legitimate public health concern, and demands a coordinated effort of the education policy makers, school administrators and teachers, community stakeholders, and parents to provide high quality PA programs in Chinese schools [20]. Nonetheless, there are many challenges and barriers in changing the daily health practices in Chinese school-age children amid recent governmental effort in improving school physical education policy [21,22]. Meanwhile, research examining the effects of modifications of school, family and community environment on physical activity and physical fitness in Chinese children remains sparse [23,24].

We conducted a clustered Randomized Controlled Trial (RCT), “The Childhood Health, Activity and Motor Performance Study (Chinese CHAMPS)”, to test the impact of a school-based intervention on the physical fitness in middle school children in China [25]. The purpose of the 4-arm study was to examine the incremental effect of adding MPA and VPA on the physical fitness among the children that were assigned to Arm 1-school physical education intervention (SPE), Arm 2-afterschool program intervention (ASP), Arm 3-SPE+ASP, and Arm 4-control condition. The study hypotheses were (1) Children in SPE, ASP and SPE+ASP would have higher levels of CRF (the primary outcome) than control children in Control; (2) Children in SPE and SPE+ASP would have higher levels of CRF than children in ASP; and (3) Children in SPE+ASP would have higher levels of CRF than children in SPE, at the end of the study. In addition, we also hypothesized that the children in the intervention arms would have more favorable changes in other physical fitness measures in children, compared to children in control at the end of the study (the secondary outcomes).

## 2. Materials and Methods

Using a clustered RCT design, Chinese CHAMPS, an 8-month multi-component physical fitness promotion intervention, was implemented in 12 middle school students from three cities in China from August 2015 to June 2016. The study protocol that described the study rationale, design, and measurement has been published previously [25]. Briefly, 12 middle schools were randomly assigned to one of four treatment conditions (one in each city): SPE intervention, ASP intervention, SPE+ASP intervention, and Control, following a 2 × 2 factorial design. The randomization was stratified by geographic location. The study participants were the 7th grade students enrolled in the study schools who were recruited with announcement posters at the schools at the beginning of the school year. The parents completed an informed consent form to permit their child’s participation in the study. All 7th grade students were eligible to participate in the study if they did not have a diagnosed physical disability and were not members of varsity sport teams. A planned sample of 540 7th grade students from 12 schools at the baseline (90% of retention at the posttest) would provide sufficient statistical power (80% power, α ≤ 0.05, 2-sided test) to test the primary study hypothesis. The Ethics Committee at the Capital University of Physical Education and Sport approved the study protocol (ChiCTR-IOR-14005388).

### 2.1. Description of the Intervention

The socio-ecological model of health promotion was the guiding philosophy of the Chinese CHAMPS that stressed the multi-level influences on a child’s PA and healthy eating. Modifications of the school policy and environment allowed the increase on the time exposed to the MVPA and VPA as well as nutrition education [26]. The goal of the intervention was to maximize the opportunities for high-intensity PA (i.e., MVPA and VPA) by the modification of school policy, an enhanced PE curriculum and a mandatory after-school PA program [27,28]. The environment for PA was modified by provision of PE equipment and teacher training that added novelty and enjoyment in children’s PA. The intervention also engaged the parents in providing a supportive environment for an active lifestyle and healthy eating at home using a mobile health-based (mHealth) campaign [29]. At the children’s level, following the competence motivation theory, the intervention activities were designed to accommodate the developmental levels of fundamental movement skills and physical fitness, and promote the sense of achievement while reducing the boredom of PA with fun and novel activities [30]. Overall, the behavioral targets of the Chinese CHAMPS are to reduce sedentary behavior, to increase the time in MVPA, and to develop the knowledge of physical fitness and nutrition in middle school students. Table 1 outlines the elements and the goals for MVPA and VPA of the Chinese CHAMPS intervention.

The design of the Chinese CHAMPS addressed two critical elements of exercise and PA interventions in children. First, past studies have not tested the feasibility and potential effect of increasing the time exposure to a large amount of MVPA and VPA that were not supported by school PE policy [31]. The PA recommendation was seldom achieved in school-based PA interventions [32,33]. Furthermore, the PA interventions have not focused on increasing VPA although many have contended its absence in the literature [27]. The Chinese CHAMPS offered the opportunity to examine the impact of the incremental amount of MVPA and VPA on the physical fitness in youth population [34,35].

Second, the motivation to participate in PA has been a serious challenge in youth population who have started the development of autonomy and sense of competence [36,37]. While younger children tend to play spontaneously with minimum adult intervention, the teenage children are less inclined to participate in activities of a high level of physical exertion and movement skills [14,38]. The decline in motivation and enjoyment as well as poor program attendance [39] were among the major barriers in promoting the MVPA in youth participants in published RCTs [40,41,42]. The Chinese CHAMPS addressed this challenge by introducing physical conditioning exercises populated by Chinese Olympic athletes. Based on the principles of the physical function training (PFT) [43], the conditioning exercises increase the fundamental movement skills and neuromuscular coordination, and target the attributes of physical fitness [44,45,46]. In contrast to traditional continuous monotonous exercises, PFT exercises are high intensity intermittent using a variety of modalities and portable equipment. Using circuit stations with small portable equipment, the PFT exercises were modified to meet the levels of children’s fundamental motor skills and physical fitness, and to challenge the children continuously as they improve. To overcome the weakness of the traditional direct skill instruction approach, the sport skills (e.g., ballgames and track and field) were taught using a game-based approach to integrate the skill instruction with active participation that were created to reduce boredom and increase time for MVPA [47,48].

With the permission and support of the local education authority and school administrators, SPE intervention modified the PE policy to offer 3 PE classes a week and daily 15-min PA-based recess to increase the amount of time for PA. This new PE policy was consistent with the recommended amount of PE time (3 h/week) for middle school students in China [49]. The PE curriculum was redesigned to provide the game-like instruction activities and innovative physical conditioning exercises that were conducive for continuous PA at a higher intensity, including a 15-min rhythmic aerobic routine for the daily recess. The time frame for the 45-min PE class was 5 min of warm-up, 12 min of instructions, 25 min of activities targeting physical fitness, and 3 min of cool-down. The modified curriculum addressed both the instructional goals and the attributes of physical fitness (CRF, strength, speed, agility, flexibility and balance). The study team developed weekly lesson plans for the PE teachers to create daily lessons. The use of portable exercise equipment allowed the teachers to vary the exercise routines while keeping the activities enjoyable and challenging [50]. Both SPE and ASP schools received the portable exercise equipment from the study. The modified PE curriculum met the national PE standards [51].

Physical fitness and nutrition education were offered to the students, using the Adolescent Fitness and Health Handbook on days that outdoor activity was not permitted due to inclement weather or air pollution. Both SPE and ASP intervention used the Handbook. The Handbook was developed by the study team and introduced age-appropriate knowledge and skills on PA and healthy eating. There were no published standards for physical fitness and nutrition education in China. As part of the physical fitness and nutrition education program, the students also received tips on PA and healthy eating twice a week on a mobile device (smart phone or tablet) delivered via a popular social media app, WeChat (https://www.wechat.com/en/).

ASP was a mandatory extracurricular activity that used the physical conditioning exercises similar to those designed for the PE classes. The time frame for the 45-min program was 5 min of warm-up, 35 min physical fitness, and 5 min of cool-down. As it was an extracurricular activity, the focus was on having fun while challenging the students physically at the end of a school day. While there was no national standard for after-school PA, the combination of SPE and ASP produced a total time exposure of 60 min of MVPA a day on school days.

Teacher training was a critical aspect of Chinese CHAMPS that offered pedagogical training on instruction design and age-appropriate teaching approaches. Using in-vivo observation and hands-on practice, the training was designed to increase the teacher’s confidence and abilities in using the redesigned curriculum activities and modifying lesson plans to meet the student needs. The teachers completed a mandatory 2-day training for SPE and 1-day training for ASP. The trained teachers and a graduate PE assistant were responsible for delivering the SPE and ASP intervention activities.

Using WeChat, the study team delivered a social media communication campaign to engage the parents of the middle school students to offer a healthy home environment [52]. The campaign sent tips to parents on supporting and encouraging physical activity, healthy eating, and sleep hygiene. The parents also received snippets on the importance and recommendations of promoting physical fitness, nutrition, sleep, reducing sedentary behavior and intake of sweetened beverages, the stages of child development, and PA recommendations for adults. The Campaign sent 82 snippets of information (2–3 messages a week). Chinese schools use WeChat as a common tool to communicate with the parents on school-related matters.

The schools in the Control condition agreed to participate in the study without receiving any intervention while conducting their PE program as usual. The PE teachers were aware that their classes were involved in a physical fitness study but did not receive any training nor made changes to the curriculum.

### 2.2. Study Measurement and Process Evaluation

The trained graduate research assistants who were blinded to the treatment conditions implemented the data collection protocol in all sites. The staff received training on the measurement procedures to standardize the protocol. The outcome measurements were conducted at the baseline and posttest of the 8-month intervention.

The physical fitness was assessed by the Chinese National Student’s Physical Fitness and Health Standard, a field-based test battery that has been validated and widely use in Chinese youth [49]. The test battery included 20MSR for CRF, broad jump for low limbs muscle strength, 1-min sit-ups for abdominal and lower back muscle strength, 50-min run for speed, T-test agility run for agility, sit and reach for flexibility, and body fat percent. The 20MSR was the primary outcome measure of the study that has been accepted globally as a proxy of CRF in the school-age population [53,54,55]. The body fat percentage was measured by a bioelectrical impedance analyzer (MC-180MA, Tanita Corporation, Tokyo, Japan) using a pediatric reference norm for Asian population. Height and weight without shoes were also measured.

The students completed several surveys in their classroom at the baseline and posttest. These include a 23-item food habits checklist [56], a 10-item nutrition knowledge test [57], sedentary behavior survey [58], and a demographic survey including their pubertal stage using the Pubertal Development Scale [59]. The parents provided their demographic information (age, gender, monthly family income, and education level) at the baseline.

Half of the study sample group were randomly chosen to wear ActiGraph GT3X-Plus accelerometers (ActiGraph, Pensacola, FL, USA) to measure their levels of physical activity and sedentary behavior at the beginning and the end of the intervention. The students wore the accelerometers on their right hips during waking hours for seven consecutive days. Data were included if an accelerometer was worn 10 h or longer a day and at least four days including one weekend day. The accelerometry data reduction followed the procedure developed for Chinese children [60]. The cut-points were 0–100, 101–2799, 2800–4000, and ≥4000 for SED, LPA, MPA and VPA, respectively [61]. ActiLife6 software (version 6.11.5, ActiGraph, Pensacola, FL, USA) was used to process the data.

The PE classes were observed to evaluate the quality of the instruction and student participation twice during the intervention (middle of Fall and Spring terms) by the PE assistant at each school. Using the System for Observing Fitness Instruction Time (SOFIT) [62], four randomly selected students were observed to estimate the amount of time that the PE teacher spent on class management and instructions of general knowledge, physical fitness, skills and games, and that the students engaged in SED, LPA, MPA and VPA in the class. In addition, 10 randomly selected students wore chest heart rate monitors (Polar Team^2^ Pro, Polar Electro Inc., Kempele, Finland) during SOFIT observation.

### 2.3. Statistical Analysis

Descriptive statistics included frequency and percentages for categorical variables, and mean with standard deviation (SD) or 95% confidence intervals (95% CI) for continuous variables. Differences in the outcome measures and characteristics of the study sample were examined using univariate F-test for continuous variables or contingency table (chi-square test or Cramer’s V) for categorical variables. Hypotheses were tested using the linear mixed model approach for repeated measures, with a compound symmetry covariance structure, and random effects for school level heterogeneity. Of primary interest was to test the three orthogonally constructed hypotheses to demonstrate the incremental effect of increased exposure on the primary and secondary outcome measures using planned contrasts. Effect size (Cohen’s *d*) was calculated for each model. Post hoc contrasts were also performed to test the difference between each of the treatment conditions with the control condition. The models were adjusted for age, sex and pubertal status, as well as within-school nesting effect. Parent education and family income were also included to explore their influences on the outcomes. Only significant variables were included in the final models. Statistical significance was set at 0.01 based on the Bonferroni adjustment for the number of comparisons, where 0.05/5 = 0.01. Analyses were performed using SAS software version 9.4 (SAS Institute Inc., Cary, NC, USA).

## 3. Results

A total of 758, 7th grade students were enrolled in the study at the baseline. Six hundred and eighty students completed both baseline and posttest assessment (89.7% retention rate) and were included in the data analysis. However, retention rate was significantly lower (*p* < 0.05) in SPE condition (79.4%) than the control condition (96.6%). Figure 1 shows the flow of the study participants.

### 3.1. Baseline Equivalence Test

The distribution of students in the four treatment conditions across the three study sites is displayed in Table 2. Male participation was higher than female across all sites. There were more students enrolled in the study site 3.

At the baseline, the students had an average age (SD) of 12.66 (0.56). The student characteristics (pubertal development, parent education levels, and family monthly income) were similar across the four treatment conditions at the baseline (see Table 3). There was no difference in student characteristics and baseline outcome measures between the students who completed both the baseline and posttest assessments and the students who were excluded from data analysis, with the exception that the students from higher family income were more likely to miss the posttest assessment (*p* < 0.001; data not shown). Finally, females were older than the males by 0.15 years and there was a gender effect on height, weight and physical fitness measures in favor of the male students.

### 3.2. Impacts on the Physical Fitness Outcomes

Table 4 shows mean change scores from baseline to posttest for each group. For the 20MSR, the mean change was 3.52 laps for the control group, 17.85 laps for the SPE group, 12.38 laps for the ASP group, and 25.78 laps for the SPE+ASP group. For the broad jump, the mean change was 5.33 cm for the control group, 21.12 cm for the SPE group, 17.19 cm for the ASP group, and 31.09 cm for the SPE+ASP group. For the 50-m run, the mean change was −0.13 s for the control group, −0.55 s for the SPE group, −0.12 s for the ASP group, and −1.07 s for the SPE+ASP group. For the sit-and-reach task, the mean change was 0.11 cm for the control group, 3.97 cm for the SPE group, 1.09 cm for the ASP group, and 6.96 cm for the SPE+ASP group. For the t test for agility, the mean change was −0.37 s for the control group, −1.22 s for the SPE group, −0.66 s for the ASP group, and −2.16 s for the SPE+ASP group. For the 1-min sit-up task, the mean change was 2.25 sit-ups for the control group, 8.16 sit-ups for the SPE group, 1.09 sit-ups for the ASP group, and 13.12 sit-ups for the SPE+ASP group. For the plank support, the mean change was −6.18 s for the control group, 29.96 s for the SPE group, −0.08 s for the ASP group, and 49.01 s for the SPE+ASP group. For body fat, the mean change was 1.3 percent for the control group, 0.8 percent for the SPE group, 0.4 percent for the ASP group, and −1.9 percent for the SPE+ASP group. Post hoc analysis revealed that these changes from baseline to posttest were significantly different between each treatment group and control, and with the exception of the changes scores for the 50-min run, sit-and-reach and plank were not significantly different between the ASP group and the control group.

In addition, Table 4 also shows the results from mixed models that tested the primary and secondary study hypotheses (i.e., the changes in the 7 test scores of physical fitness between the baseline and posttest). 

**Hypothesis** **1.**
*In terms of contrasting the pooled effect of SPE, ASP, and SPE+ASP vs. Control.*


**Hypothesis** **2.**
*the pooled effect of SPE and SPE+ASP vs. ASP only.*


**Hypothesis** **3.**
*the effect of SPE+ASP vs. ASP on the 20MSR scores were all statistically significant after adjusting for covariates.*


The pooled effect of SPE, ASP, and SPE+ASP resulted in a mean change of 15.2 more laps than the control group for the 20 MSR (*p* < 0.001). The pooled effect of SPE and SPE+ASP resulted in a mean change of 9.4 more laps than the ASP only group (*p* < 0.001). The effect of SPE+ASP resulted in a mean change of 13.6 more laps than the ASP only group (*p* < 0.001). The pooled effect of SPE, ASP, and SPE+ASP resulted in a mean change of 17.0 cm more than the control group for the broad jump (*p* < 0.001). The pooled effect of SPE and SPE+ASP resulted in a mean change of 9.7 cm more than the ASP only group (*p* < 0.001). The effect of SPE+ASP resulted in a mean change of 15.0 cm more than the ASP only group (*p* < 0.001). The pooled effect of SPE, ASP, and SPE+ASP resulted in a mean change of −0.4 s less than the control group in the 50-m run (*p* < 0.001). The pooled effect of SPE and SPE+ASP resulted in a mean change of −0.7 s less than the ASP only group (*p* < 0.001). The effect of SPE+ASP resulted in a mean change of −1.0 s less than the ASP only group (*p* < 0.001). The pooled effect of SPE, ASP, and SPE+ASP resulted in a mean change of 3.5 cm more than the control group in the sit-and-reach (*p* < 0.001). The pooled effect of SPE and SPE+ASP resulted in a mean change of 4.1 cm more than the ASP only group (*p* < 0.001). The effect of SPE+ASP resulted in a mean change of 5.7 cm more than the ASP only group (*p* < 0.001). The pooled effect of SPE, ASP, and SPE+ASP resulted in a mean change of −1.0 s less than the control group for the t test of agility (*p* < 0.001). The pooled effect of SPE and SPE+ASP resulted in a mean change of −1.0 s less than the ASP only group (*p* < 0.001). The effect of SPE+ASP resulted in a mean change of −1.5 s less than the ASP only group (*p* < 0.001). The pooled effect of SPE, ASP, and SPE+ASP resulted in a mean change of 5.1 sit-ups than the control group (*p* < 0.001). The pooled effect of SPE and SPE+ASP resulted in a mean change of 9.5 more sit-ups than the ASP only group (*p* < 0.001). The effect of SPE+ASP resulted in a mean change of 12.0 more sit-ups than the ASP only group (*p* < 0.001). The pooled effect of SPE, ASP, and SPE+ASP resulted in a mean change of 31.8 s more than the control group for the plank (*p* < 0.001). The pooled effect of SPE and SPE+ASP resulted in a mean change of 37.2 s more than the ASP only group (*p* < 0.001). The effect of SPE+ASP resulted in a mean change of 48.6 s more than the ASP only group (*p* < 0.001). The pooled effect of SPE, ASP, and SPE+ASP resulted in a mean change of −1.6 percent less body fat than the control group (*p* < 0.001). The pooled effect of SPE and SPE+ASP resulted in a mean change of −1.1 percent less than the ASP only group (*p* < 0.001). The effect of SPE+ASP resulted in a mean change of −2.4 percent less than the ASP only group (*p* < 0.001).

### 3.3. Process Evaluation of the Intervention Implementation

Process-related information was evaluated to examine the effects on the targeted behaviors in response to the implementation of incremental changes in school policy and environmental modifications. An examination of a student’s scores on nutrition knowledge test and food habits showed improvement from baseline to posttest from three intervention groups compared to the students in control (see Table 5). The self-reported time on sedentary activities from the three intervention groups also went down consistently during weekdays and to a lesser extent during the weekend days compared to the control group at the posttest. Table 6 shows the percent of time estimated from accelerometry that the students spent in SED, LPA, MPA, VPA, and MVPA during the past 7 days, the school days, and weekend days at the baseline and posttest. The students in ASP, SPE and SPE+ASP condition increased their time in MPA, VPA, and MVPA, which decreased in control students during the school days from the baseline to the posttest. The time in SED and LPA increased in all students during the school days over the study period. On the weekend days, there were no discernable changes in MVPA in all students from the baseline to the posttest. Observation of PE classes by SOFIT also demonstrated that the students in SPE and SPE+ASP condition increased their time in VPA and MVPA, and showed increases in the percentage of HR in ≥130 bpm, ≥140 bpm, and ≥150 bpm during PE classes while minimum change was observed in ASP and control students at the posttest (see Table 6). Finally, the amount of time that the intervention teachers spent in class management decreased and the time in instruction-related activities increased at the posttest observation.

## 4. Discussion

Chinese CHAMPS is effective in enhancing the levels of physical fitness with similar impacts on the primary and secondary outcome measures in Chinese middle school students. In supporting the study hypotheses, the finding indicates a robust incremental effect of the intervention with the most improvement in physical fitness in SPE+ASP, and followed by SPE, and ASP only intervention. Furthermore, SPE was superior to ASP. Previous RCTs have demonstrated that modification of PE curriculum alone does not always generate a sufficient amount of MVPA and improve physical fitness and health outcomes [63,64,65], and school-based PA interventions have produced a small effect in increasing the duration of MVPA and rate of meeting daily PA recommendations [27,32].

The incremental effect of Chinese CHAMPS on physical fitness is consistent with systematic reviews of school-based RCTs that favored multi-component interventions in various regions across the globe [66,67,68,69,70], amidst the methodological weaknesses and small effect size (e.g., inadequate dose and intensity of PA and poor fidelity of the intervention implementation) [27,33,71]. Numerous studies have demonstrated that increasing PE class frequency and duration led to significant improvement in multiple attributes of physical fitness in elementary school children [72,73,74]. Current evidence points to the necessity of allocation of extra time to PA, in addition to enhancement in PE curriculum, to increase the compliance to the recommended daily physical activity and maximize the intervention effect in school-settings [34,73]. For example, Bassett et al. estimated that modification of school PA policy on PE time, recess, and after-school programs can generate up to 38 additional minutes of MVPA in school age children. A meta-analysis of 14 RCTs of school-based interventions reported a 24% increase in MVPA time during PE classes [75]. A Cochrane review of 26 school-based PA intervention studies found that multicomponent interventions increased time in MVPA and decreased time in TV watching during school days [27]. As shown by the accelerometry data, there were improvements in the levels of SED and PA at the posttest compared to those reported in observation studies of similar age children from China [10] and other countries [76,77,78,79]. The percent of time in MPA, VPA, and MVPA doubled or tripled in the students in the three intervention conditions at the end of the intervention. Meanwhile, the students in the control condition decreased their time in MVPA and VPA at the end of the intervention. Thus, school policy that addresses PE classes, recesses, and after-school activities is instrumental in augmenting PA and improving physical fitness in school-age children.

The emphasis on high-intensity PA over an 8-month period in Chinese CHAMPS was likely the reason for the improvement of CRF, compared to past studies that did not focus on high intensity PA for a longer period of time [80]. For example, one early Danish study examined the effect of adding three 50-min physical training sessions each week that targeted high intensity PA (70% of maximum heart rate) for eight months [74]. While no effect was observed at 3-month, the intervention led to a significant increase in maximum oxygen uptake in normative and hypertensive 9–11 year-old children, compared to hypertensive controls at 8-month posttest. In the present study, the heart rates of the students in SPE intervention were ≥140 bpm and ≥150 bpm for >60% and >25% of the time in PE classes, respectively, at the end of the intervention. This information suggested that the students spent a significant amount of time exercising at an intensity around the lower boundary of VPA that were expected in high-intensity physical training programs in children [81]. Few published studies examined the effect of school-based PA interventions that targeted VPA in middle school children. Given the limited time allocated for PA in schools, school-based PA programs can be strengthened with high intensity aerobic activities (70–85% of maximum heart rate) that are more effective and efficient to improve CRF [34]. Chinese CHAMPS demonstrates a promising physiological approach to promote physical fitness in school-age children using vigorous forms of PA.

Built on earlier evidence that promoting active learning (e.g., fitness infusion and game-based approaches) can increase MVPA and participation motivation [75,82], Chinese CHAMPS was designed to target all attributes of physical fitness and fundamental movement skills by the use of innovative physical conditioning exercises based on PFT [44] or INT [83] and game-based instruction approach. Although relatively new in school-based PA interventions [84], there are reports of promising effects of PFT-based exercise programs on a variety of physical fitness measures in elementary school children in the United States [83,85]. On the other hand, there is strong evidence that a game-based instruction approach can increase engagement and reduce boredom in participation of PA in children [36,82]. Motivational and attendance issues have been reported in high-intensity PA programs that are not enjoyable and developmentally appropriate [30,42,86]. The unique combination of PFT-based exercises and game-based approach make Chinese CHAMPS as a likely contributing factor for the success of the intervention and should be further examined in future studies.

It is well recognized that the after-school period is spent predominantly in sedentary activities [58,87,88]. Increasing MVPA during the after-school period has great potential to increase the compliance with PA recommendation in children and reduce sedentary behavior [79,89], and ameliorate the impact of the obesogenic environment on health-related outcomes in children [90,91]. Findings from Chinese CHAMPS show that adding a quality afternoon PA program alone can be efficient and effective to produce the dose of PA that is sufficient to improve physical fitness, and increases the likelihood of meeting the recommended physical activity daily. Although the practice of offering organized intramural PA is not new in Chinese schools, Chinese CHAMPS is among the first to provide evidence that high-quality after-school programs can increase MVPA and VPA and physical fitness in Chinese students. A recent study found that offering after-school PA programs was associated with improved physical fitness in Taiwanese children [92]. Therefore, findings from this study add support to an emerging literature on after-school PA interventions [93].

Finally, the evaluation of the process-related information indicated a high fidelity of the implementation of the intervention protocol that is comparable or better than past studies of school-based PA interventions [50,75,94]. The aggregates of evaluation information (accelerometry, SOFIT, and heart rates) demonstrated changes in the targeted behaviors that corresponded to the nature and the type of the intervention treatment. For example, the increases in the portion of active time as well as activity intensity, and improving the teacher’s effectiveness in delivering quality instruction were only observed in students receiving SPE intervention.

### Study Strengths and Limitations

A major strength of the study was the level of support and participation by the study schools that accepted the random assignment and implemented the policy changes as proposed. Another strength was the process evaluation that used both self-report and objective measures to demonstrate the fidelity of the intervention implementation. Lastly, the Chinese CHAMPS can be scaled for translation since the modified curriculum is based on the national physical education standards, and existing infrastructure in most urban Chinese schools.

There were several weaknesses in the study. First, there was no follow-up to examine if the changes in the physical fitness were sustainable beyond the 8-month intervention [27]. Second, accelerometry data were collected from a portion of the study sample. Therefore, we were not able to examine the dose-effect relation between the PA and physical fitness. Third, we did not collect information on school attendance and attendance in the after-school program. Past studies have linked poor attendance in after-school programs to reduced intervention effects [95]. Chinese schools have a very stringent attendance policy and as a result, students seldom miss school. Therefore, attendance is not likely to attenuate the intervention effect in this study. Fourth, we did not collect information on PA-related motivation and enjoyment from the students which were the behavioral mediators targeted by the intervention. As a result, we could not establish to what extent the student’s motivation might have contributed to the outcomes of the intervention. Previous studies have linked a higher level of motivation to a child’s improvement in CRF and MVPA [96,97]. Similarly, we did collect information to evaluate the effectiveness of the health education delivered to the students during inclement weather days. However, there were clear increases in a student’s nutritional knowledge and nutritional habit scores, and reduction in a student’s self-report of sedentary behavior at the posttest. Finally, a social media campaign was used to engage the parents in supporting their children to be active and eating healthy at home. However, we did not collect information to assess the feasibility and effectiveness of the parent engagement program. Mobile health intervention designed for parents to improve their child’s physical activity and dietary behavior is still in the early stage of development. Previous studies also found that a higher level of parental engagement is associated with improvement in a child’s health behaviors [52,98]. Future studies should examine the acceptability and feasibility of the parent engagement program implemented in Chinese CHAMPS.

Current school policy on PE and PA time in Chinese schools mandates three 45-min PE classes a week and 25–30 min of PA-oriented daily recess in middle schools [99]. Findings from this study suggest that the current policy can be effective in improving the physical fitness in Chinese children if fully implemented. However, to achieve the recommended PA requires the schools to allocate PA time during after-school hours, to institutionalize the time requirement for PE class and recess into the school routine, and to enhance the quality of PE curriculum, exercise equipment, and teacher training. An emerging body of literature shows that improvement in physical fitness can lead to an increase in cognitive functions in school-age children [100]. Education policy-makers and school officials should take advantage of the health and cognitive benefits in promoting and reinforcing the recommended PA in schools.

## 5. Conclusions

Findings of the Chinese CHAMPS demonstrate that multi-level school-based intervention, based on a socio-ecological model of health promotion, is efficacious in improving CRF and other attributes of physical fitness in middle school children. In supporting the study hypotheses, the Chinese CHAMPS illustrated an incremental effect on multiple attributes of physical fitness in middle school students, and provided the much-needed evidence to increase the opportunity for PA in schools and to introduce high-intensity exercises in school-based PA programs in Chinese schools. The rigor of the findings in this study was strengthened by favorable changes in objectively and subjectively measured behaviors and teacher practices. Future studies should replicate the study to examine the influence of a student’s motivation and parental engagement in the improvement of physical fitness, and to test the effectiveness of Chinese CHAMPS using the framework of the implementation science [101].

## Figures and Tables

**Figure 1 ijerph-16-04412-f001:**
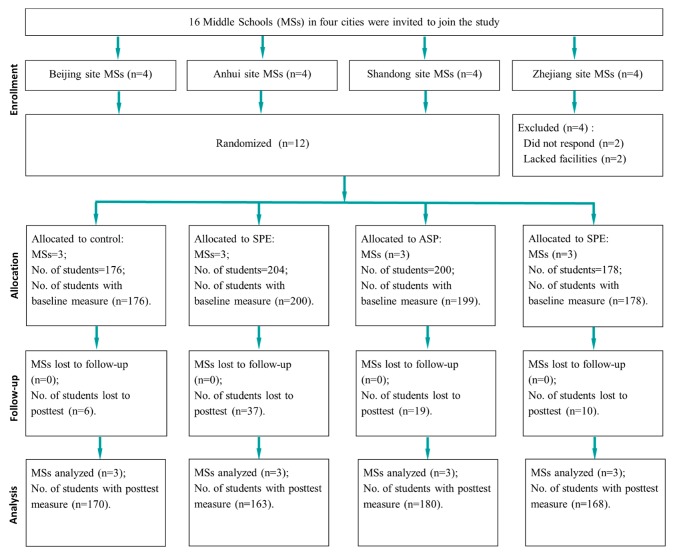
Participant Flow.

**Table 1 ijerph-16-04412-t001:** Outlines of Chinese CHAMPS Intervention Design: Intervention Elements and Activity Goals †.

Intervention Treatment		Offering the After-School Program	
		No	Yes
**Modifying school physical education program**	No	No change in PE policy No teacher training No after-school program	Provision of after-school PA program (adding 45 min of MVPA 2 d/week) Provision of portable exercise equipment Fitness & nutrition education Bi-weekly text messages to students on healthy tips No change in PE policy Parent engagement
		90 min MVPA/week (45 min PE × 2 d/week)	180 min MVPA/week (45 min PE × 2 d/week & 45 min PA × 2 d/week); ≥45 min VPA/week
	Yes	PE policy change (3 PE classes/week and daily recess) 2-d teacher training Provision of portable exercise equipment PE curriculum re-design Fitness & nutrition education Bi-weekly text messages to students on healthy tips Parent engagement	PE policy change (3 PE classes/week and daily recess) Provision of after-school PA program (adding 45 min of MVPA 2 d/week) 3-d teacher training Provision of portable exercise equipment PE curriculum re-design Physical fitness & nutrition education Bi-weekly text messages to students on healthy tips Parent engagement
		210 min MVPA/week (45 min PE × 3 d/week & 15 min recess × 5 d/week); ≥105 min VPA/week	300 min MVPA/week (45 min PE × 3 d/week, 15 min recess × 5 d/week & 45 min PA × 2 d/week); ≥150 min VPA/week

† For the time period 8:00 a.m. to 5 p.m. that students stayed on campus. PA, physical activity; Min, minutes; d, day; PE, physical education; MVPA, moderate to vigorous physical activity.

**Table 2 ijerph-16-04412-t002:** Distribution of Students across Study Sites (*n*, %).

Study Sites	Control			School Physical Education Intervention			After-School Program Intervention			School Physical Education & After-School Intervention			All Sites		
	M	F	*n*	M	F	*n*	M	F	*n*	M	F	*n*	M	F	*n*
Site 1	31, 63%	18, 37%	49	21, 57%	16, 43%	37	23, 49%	24, 51%	47	33, 62%	20, 38%	53	108, 58%	78, 42%	186
Site 2	28, 53%	25, 47%	53	26, 50%	26, 50%	52	19, 54%	16, 46%	35	24, 49%	25, 51%	49	97, 51%	92, 49%	189
Site 3	36, 53%	32, 47%	68	38, 52%	35, 48%	73	54, 55%	44, 45%	98	30, 45%	36, 55%	66	158, 52%	147, 48%	305
All sites	95, 59%	75, 41%	170	85, 52%	77, 48%	162	96, 53%	84, 47%	180	87, 52%	81, 48%	168	363, 53%	317, 47%	680

M, Males; F, Females.

**Table 3 ijerph-16-04412-t003:** Study Participant Characteristics at the Baseline.

Variables		All (*n* = 680)	Control (*n* = 170)	SPE (*n* = 162)	ASP (*n* = 180)	SPE+ASP (*n* = 168)	Group Comparisons (*p* < 0.05) †
Age, mean (SD) (y)		12.66 (0.56)	12.68 (0.63)	12.69 (0.53)	12.67 (−0.53)	12.63 (0.54)	F > M
Female, No. (%)		317 (46.6)	75 (44.1)	77 (47.5)	84 (46.7)	81 (48.2)	M > F
Pubertal stage, No. (%) ‡	2	219 (32.2)	49 (28.8)	52 (32.1)	68 (37.8)	50 (29.8)	
	3	327 (48.1)	86 (50.6)	80 (49.4)	81 (45.0)	80 (47.6)	
	4	134 (19.7)	35 (20.6)	30 (18.5)	31 (17.2)	38 (22.6)	
Father education level, No. (%)	Junior high school	143 (21.0)	40 (23.5)	50 (30.9)	15 (8.3)	38 (22.6)	
	High school	204 (30.0)	60 (35.3)	54 (33.3)	46 (25.6)	44 (26.2)	
	Junior Colleges	145 (21.3)	40 (23.5)	23 (14.2)	47 (26.1)	35 (20.8)	
	University	114 (16.8)	22 (12.9)	18 (11.1)	44 (24.4)	30 (17.9)	
	Postgraduate	42 (6.2)	3 (1.8)	3 (1.9)	21 (11.7)	15 (8.9)	
Mother education level, No. (%)	Junior high school	180 (26.5)	51 (30.0)	61 (37.7)	18 (10.0)	50 (29.8)	
	High school	209 (30.7)	63 (37.1)	56 (34.6)	52 (28.9)	38 (22.6)	
	Junior college	146 (21.5)	28 (16.5)	27 (16.7)	56 (31.1)	35 (20.8)	
	University	95 (14.0)	20 (11.8)	6 (3.7)	39 (21.7)	30 (17.9)	
	Postgraduate	28 (4.1)	5 (2.9)	3 (1.9)	10 (5.6)	10 (6.0)	
Monthly family income, No. (%)	RMB ≤ 5000	183 (26.9)	49 (28.8)	67 (41.4)	27 (15.0)	40 (23.8)	
	RMB 5000-10000	301 (44.3)	67 (39.4)	66 (40.7)	87 (48.3)	81 (48.2)	
	RMB 10001-15000	111 (16.3)	35 (20.6)	14 (8.6)	43 (23.9)	19 (11.3)	
	RMB ≥ 15000	63 (9.3)	16 (9.4)	6 (3.7)	18 (10.0)	23 (13.7)	
Height, mean (SD) (cm)		162.24 (−6.59)	161.93 (−6.91)	162.09 (−6.69)	162.53 (−6.53)	162.39 (−6.29)	M > F
Weight (kg)		52.73 (11.17)	51.40 (10.76)	53.09 (11.38)	52.82 (9.91)	53.65 (12.60)	M > F
20-m shuttle run (laps)		38.33 (15.25)	38.09 (14.59)	35.96 (14.13)	40.07 (16.18)	39.00 (15.75)	M > F
Broad jump (cm)		178.69 (23.30)	179.75 (21.00)	177.96 (23.25)	179.44 (22.81)	177.51 (26.06)	M > F
50-m run (seconds)		8.76 (0.95)	8.85 (1.09)	8.78 (0.88)	8.64 (0.80)	8.80 (1.00)	F > M
Sit-and-reach (cm)		9.50 (5.64)	9.89 (5.48)	9.28 (5.12)	9.39 (6.56)	9.46 (5.25)	F > M
T test for agility (seconds)		12.40 (1.31)	12.54 (1.37)	12.40 (1.43)	12.22 (1.15)	12.45 (1.29)	F > M
1-min sit-ups (counts)		36.91 (8.93)	36.23 (9.73)	35.73 (9.20)	38.43 (8.34)	37.09 (8.24)	M > F
Plank support (seconds)	34	100.08 (39.38)	98.74 (41.10)	97.35 (37.59)	100.21 (36.81)	103.92 (41.91)	F > M
Body fat percent (%)		20.96 (5.68)	20.92 (5.46)	21.10 (5.70)	20.84 (4.99)	20.93 (6.42)	F > M

† Group comparison by treatment conditions and gender was tested using univariate F-test for continuous variables or contingency table (ꭕ2 or Cramer’s V) for categorical variables. ‡ Pubertal stage 1 and 5 were combined with its nearest stage due to small frequency of occurrence Control, control condition; SPE, school physical education intervention; ASP, after-school program intervention; SPE+ASP, SPE and ASP combined intervention.

**Table 4 ijerph-16-04412-t004:** Mean Changes of Physical Fitness Measures between Baseline and Posttest and Planned Contrasts †.

Outcome	Estimated Mean (95% CI)				Contrasts Coefficient (95% CI); Effect Size *; *p* Value		
Measure	Control	SPE	ASP	SPE+ASP	SPE & ASP & SPE+ASP vs. Control	SPE & SPE+ASP vs. ASP	SPE+ASP vs. ASP
20-m shuttle run (laps)	3.52 ^a, b, c^ (1.18, 5.85)	17.85 ^a^ (15.68, 20.02)	12.38 ^b^ (10.2, 14.56)	25.78 ^c^ (23.7, 27.86)	15.2 (12.3, 18.2); 0.29; <0.001	9.4 (6.5, 12.4); 0.18; <0.001	13.6 (10.2, 17.0); 0.22; <0.001
Broad jump (cm)	5.33 ^a, b, c^ (1.94, 8.71)	21.12 ^a^ (18.1, 24.14)	17.19 ^b^ (14.13, 20.24)	31.09 ^c^ (27.6, 34.58)	17.0 (12.8, 21.3); 0.22; <0.001	9.7 (5.3, 14.2); 0.12; <0.001	15.0 (9.9, 20.0); 0.17; <0.001
50-m run (seconds)	−0.13 ^a, b^ (−0.27, 0.00)	−0.55 ^a^ (−0.66, −0.43)	−0.12 (−0.26, 0.01)	−1.07 ^b^ (−1.21, −0.93)	−0.4 (−0.6, −0.3); 0.14; <0.001	−0.7 (−0.9, −0.5); 0.23; <0.001	−1.0 (−1.2, −0.8); 0.29; <0.001
Sit-and-reach (cm)	0.11 ^a, b^ (−0.57, 0.78)	3.97 ^a^ (3.28, 4.66)	1.09 (0.36, 1.82)	6.96 ^b^ (6.16, 7.75)	3.5 (2.5, 4.5); 0.20; <0.001	4.1 (3.1, 5.1); 0.23; <0.001	5.7 (4.6, 6.8); 0.28; <0.001
T test for agility (seconds)	−0.37 ^a, b, c^ (−0.6, −0.14)	−1.22 ^a^ (−1.43, −1.01)	−0.66 ^b^ (−0.84, −0.48)	−2.16 ^c^ (−2.37, −1.96)	−1.0 (−1.2, −0.7); 0.20; <0.001	−1.0 (−1.3, −0.7); 0.21; <0.001	−1.5 (−1.8, −1.2); 0.28; <0.001
1-min sit-ups (counts)	2.25 ^a, b, c^ (0.95, 3.55)	8.16 ^a^ (6.83, 9.48)	1.09 ^b^ (−0.19, 2.37)	13.12 ^c^ (11.85, 14.38)	5.1 (3.3, 6.9); 0.16; <0.001	9.5 (7.8, 11.3); 0.30; <0.001	12.0 (10.0, 14.0); 0.34; <0.001
Plank support (seconds)	−6.18 ^a, b^ (−13.01, 0.66)	29.96 ^a^ (21.26, 38.65)	−0.08 (−7.34, 7.18)	49.01 ^b^ (42.71, 55.31)	31.8 (22.4, 41.2); 0.19; <0.001	37.2 (27.8, 46.7); 0.22; <0.001	48.6 (37.8, 59.5); 0.25; <0.001
Body fat (percent)	1.3 ^a, b, c^ (0.7, 2.0)	0.8 ^a^ (0.3, 1.3)	0.4 ^b^ (−0.3, 1.2)	−1.9 ^c^ (−2.7, −1.1)	−1.6 (−2.4, −0.8); 0.12; <0.001	−1.1 (−2.0, −0.1); 0.07; =0.03	−2.4 (−3.5, −1.4); 0.14; <0.001

† Models adjusted for child’s age, sex, pubertal stage, father’s education level, mother’s education level, family income per month. Control, control condition; SPE, school physical education intervention; ASP, after-school program intervention; SPE+ASP, SPE and ASP combined intervention. * Effect size measured as Cohen’s D. ^a, b, c^ Same letter denotes significant difference between the two treatment conditions in unadjusted post hoc test (*p* < 0.05 with Bonferroni correction).

**Table 5 ijerph-16-04412-t005:** Student’s Nutrition Knowledge and Food Habits (Means and SD) and Self-Reported Sedentary Behavior (%) at Baseline and Posttest.

Test Items or Variables	Control (*n* = 164)		SPE (*n* = 160)		ASP (*n* = 171)		SPE+ASP (*n* = 163)	
	Baseline	Posttest	Baseline	Posttest	Baseline	Posttest	Baseline	Posttest
Nutrition Knowledge Test *	4.37 1.21	5.28 1.40 †	4.34 1.25	7.01 1.65 †	4.33 1.40	7.17 1.97 †	4.37 1.41	7.64 1.70 †
Food Habits Checklist **	12.15 2.47	13.80 2.59 †	12.41 2.69	17.12 3.00 †	12.35 1.90	17.74 2.66 †	12.21 2.57	18.05 3.03 †
Time spent on TV watching during school days	19.97	20.23	22.85	17.90	20.38	14.49	18.37	12.61
	1.80	1.78	2.06	1.51 †	1.78	1.45 †	1.94	1.38 †
Time spent on TV watching during weekend days	65.37	61.92	64.39	63.04	51.77	52.35	60.06	60.39
	5.86	4.88	4.57	4.57	4.51	4.64	5.61	5.22
Time spent on internet during school days	23.86	24.66	27.82	24.50	27.40	16.21	25.31	15.98
	2.37	2.22	2.56	1.98	2.48	1.57 †	2.52	1.45 †
Time spent on internet during weekend days	59.63	55.74	68.02	62.81	53.55	50.94	56.98	55.04
	5.03	4.20	5.73	5.33	4.93	4.60	5.60	5.19
Time spent on computer games during school days	9.26	10.23	10.82	8.29	8.21	5.76	8.37	5.34
	1.53	1.68	1.73	1.36 †	1.60	1.22 †	1.75	1.34 †
Time spent on computer games during weekend days	37.94	33.37	34.33	38.30	30.96	29.80	29.90	30.24
	4.41	4.03	3.42	3.70	4.03	3.86	3.90	4.18

Control, control condition; SPE, school physical education intervention; ASP, after-school program intervention; SPE+ASP, SPE and ASP combined intervention. * Higher scores indicate higher level of nutrition knowledge. ** Higher scores indicated higher level of preference for nutritious foods and beverages. † Significant change from baseline to posttest (Paired-sample T-test, *p* < 0.05 with Bonferroni correction).

**Table 6 ijerph-16-04412-t006:** Process Information Related to Implementation of Physical Activity Components of CHAMPS Based on Accelerometry, Heart Rate Monitoring, and Observation of Physical Education Classes at Baseline and Posttest.

Variables	Control		SPE		ASP		SPE+ASP	
	Baseline	Posttest	Baseline	Posttest	Baseline	Posttest	Baseline	Posttest
Daily wearing time for the past 7 days (min/day)	818.11	803.95	841.55	834.84	818.48	765.23	842.01	760.83
	146.04	186.08	179.19	185.59	159.66	221.05	155.19	161.44 †
% of time in sedentary behaviors per day for the past 7 days	71.44	74.40	70.10	75.51	70.38	75.74	69.88	74.51
	9.11	10.80	8.53	9.64 †	8.95	8.81 †	10.28	11.95
% of time in LPA per day for the past 7 days	26.09	23.36	27.22	19.14	27.06	20.04	27.54	18.28
	8.58	9.80	7.93	9.37 †	8.24	8.84 †	9.43	11.95 †
% of time in MPA per day for the past 7 days	1.54	1.36	1.70	2.73	1.54	2.33	1.59	3.65
	0.97	0.90	0.85	1.15 †	0.74	0.83 †	1.03	0.98 †
% of time in VPA per day for the past 7 days	0.93	0.87	0.98	2.62	1.02	1.89	0.99	3.57
	0.64	0.77	0.67	0.86 †	0.71	0.75 †	0.75	1.00 †
% of time in MVPA per day for the past 7 days	2.47	2.23	2.68	5.35	2.56	4.22	2.58	7.21
	1.43	1.52	1.42	1.79 †	1.27	1.39 †	1.71	1.84 †
Daily wearing time during weekdays (min/day)	825.95	832.14	838.04	882.40	820.14	800.05	834.74	813.05
	149.01	181.95	183.02	226.39	170.38	240.31	166.63	191.16
% of time in sedentary behaviors per day during weekdays	69.61	71.65	69.35	74.66	70.05	73.79	69.17	72.94
	9.45	10.96	8.97	9.23 †	8.61	9.27	11.27	11.39
% of time in LPA per day during weekdays	27.53	25.97	27.53	18.87	27.12	21.34	28.02	18.29
	8.76	9.99	8.23	9.35 †	7.85	9.40 †	10.23	11.72 †
% of time in MPA per day during weekdays	1.75	1.46	1.96	3.24	1.62	2.69	1.68	4.38
	1.07	0.94	1.06	1.31 †	0.73	0.99 †	1.05	1.29 †
% of time in VPA per day during weekdays	1.10	0.92	1.15	3.23	1.21	2.18	1.13	4.38
	0.76	0.78	0.83	1.10 †	0.90	0.83 †	0.82	1.26 †
% of time in MVPA per day during weekdays	2.85	2.38	3.12	6.47	2.83	4.87	2.81	8.76
	1.64	1.56	1.74	2.14 †	1.46	1.61 †	1.81	2.39 †
Daily wearing time during the weekend (min/day)	836.83	765.18	899.20	757.69	835.11	716.60	903.35	668.29
	254.07	279.16	397.15	403.32	300.46	351.95	319.95	243.45 †
% of time in sedentary behaviors per day during the weekend	77.81	82.69	72.44	77.50	70.98	81.14	72.29	79.74
	14.84	14.97	12.95	19.32	17.23	16.52 †	15.95	17.82 †
% of time in LPA per day during the weekend	20.95	15.58	26.07	20.68	27.30	17.02	25.80	18.06
	14.04	13.53	12.25	17.66	16.28	15.12 †	14.81	16.02 †
% of time in MPA per day during the weekend	0.84	1.07	0.99	1.18	1.26	1.01	1.32	1.27
	1.22	1.28	0.96	1.74	1.25	1.32	1.58	1.84
% of time in VPA per day during the weekend	0.40	0.67	0.50	0.64	0.47	0.93	0.58	0.93
	0.61	1.07	0.72	1.17	0.59	1.47	0.86	1.81
% of time in MVPA per day during the weekend	1.24	1.73	1.49	1.82	1.73	1.94	1.91	2.20
	1.66	2.18	1.45	2.75	1.71	2.57	2.25	3.51
Average heart rate during PE class (beats/minute)	103.02	106.33	102.43	141.29	101.31	107.43	101.86	142.18
	15.11	9.58	14.55	7.70 †	13.37	17.57 †	11.54	10.40 †
Average of maximum heart rate during PE class (beats/minute)	122.21	119.44	120.51	152.85	118.89	125.46	121.63	156.48
	14.81	9.98	15.03	7.42 †	13.09	21.00	13.49	9.03 †
% ≥130 bpm of HR during PE class	14.82	12.12	14.32	72.23	12.17	13.01	13.06	72.73
	15.32	3.10	14.72	6.43 †	13.03	9.63	13.01	9.55 †
% ≥140 bpm of HR during PE class	6.88	4.04	7.26	60.72	5.87	5.71	6.41	61.75
	14.45	2.48	14.40	3.79 †	12.06	8.74 †	12.06	4.81 †
% ≥150 bpm of HR during PE class	1.70	0.00	2.00	5.71	1.27	1.39	1.22	25.56
	6.28	0.00	6.61	8.74 †	5.67	4.56	5.25	2.49 †
Walking during PE class based on SOFIT (min)	12.42	9.49	12.24	21.50	12.47	9.95	12.83	22.14
	3.65	3.49	3.23	4.09 †	3.79	3.05	3.83	4.16 †
VPA during PE class based on SOFIT (min)	2.80	3.50	3.82	8.97	3.61	4.21	2.83	8.42
	2.72	2.93	2.85	3.01 †	3.05	2.94	2.48	3.01 †
MVPA during PE based on SOFIT (min)	15.22	12.99	16.06	30.47	16.08	14.16	15.66	30.56
	3.64	4.23	2.57	3.43 †	3.45	2.90	3.53	3.48 †
Class time in class management during PE class based on SOFIT (min)	13.19	11.76	13.12	4.80	13.36	13.72	12.87	4.70
	3.38	5.00	3.76	1.63 †	3.82	4.19	2.94	1.85 †
Class time in knowledge instruction during PE class based on SOFIT (min)	11.11	11.29	11.12	4.71	11.14	9.40	10.01	4.45
	5.00	4.01	5.02	2.12 †	4.79	4.07	4.78	2.06 †
Class time in physical fitness instruction during PE class based on SOFIT (min)	7.55	7.39	7.17	19.79	7.42	7.17	7.48	19.16
	3.86	3.33	3.71	4.33 †	3.61	3.38	4.15	4.55 †
Class time in skill instruction during PE class based on SOFIT (min)	5.12	4.73	5.51	10.17	5.31	4.80	5.72	11.11
	1.92	2.09	2.59	3.46 †	2.26	1.88	2.47	2.98 †
Class time in game instruction during PE class based on SOFIT (min)	5.28	5.59	5.72	3.71	5.44	5.62	6.29	3.45
	2.95	2.21	2.62	1.51	2.34	2.02	3.03	1.34 †

Control, control condition; SPE, school physical education intervention; ASP, after-school program intervention; SPE+ASP, SPE and ASP combined intervention; VPA, vigorous physical activity; MVPA, moderate to vigorous physical activity; PE, physical education; SOFIT, the System for Observing Fitness Instruction Time; † Significant change from baseline to posttest (Paired-sample T-test, *p* < 0.05 with Bonferroni correction).

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
