# Peer review of "Impact on Physical Fitness of the Chinese CHAMPS: A Clustered Randomized Controlled Trial"

_ijerph, 2019, doi:10.3390/ijerph16224412_

Round 1

Reviewer 1 Report

Thank you for inviting me to review the paper on “Impact on physical fitness of the Chinese CHAMPS: a clustered randomized controlled trial”. This research is an example of the design of a good multi-component program for the promotion and improvement of health in schools.This is a good paper and deserves to be published in IJERPH.

The introduction clearly presents the research problem and the proposed objectives are adequate and subsequently well answered in the discussion.

The methodology is adequate and is supported by the previous publication of its protocol. The results provide the necessary data to respond appropriately to the objectives and use appropriate statistical methods.

The main criticism of this paper is that it is perhaps too ambitious in presenting results. There are many variables treated and difficult to follow for the reader.

In addiction, this reviewer has the following recommendations:

ABSTRACT:

It is to consider positively the registry of the present investigation in Chinese Clinical Trial Registry. Could it also have been carried out in the International Clinical Trials Registry Platform?

Obviously before a study with such a multitude of variables it is necessary to provide many details even in the summary. Therefore, perhaps the summary is somewhat more extensive than desired.

Reformulate some keywords keywords (i.e. Physical function training program; Multi-component promotion; School physical education policy). Some of them would not really help in the search processes in the databases. Try to make them clearer and more direct.

INTRODUCTION

Sometimes the acronym appears for the first time without detailing the corresponding word, i. e. line 66 RCT y line 74 CRF

METHODS

Perhaps the characteristics of the intervention are addressed in too much detail. Even more when there is a previous publication precisely on this protocol.

In the opinion of this reviewer, the information and variables presented for a single article are too many. (Fitness, measure of levels of physical activity and sedentary behavior at the beginning and the end of the intervention, observation of physical education clases, food habits checklist, nutrition knowledge test, sedentary, rate monitors, behavior survey, demographic survey including their pubertal stage, development Scale, parents demographic information, etc).

It might have been easier to present it in two different publications.

RESULTS

In table 2, put down the meaning of M and F. We assume that it is Males and Females but it would be appropriate to detail these issues at the bottom of the table.

In Table 5, Student’s Nutrition Knowledge and Food Habits and Self-Reported Sedentary Behavior at Baseline and Posttest can be observed. Pre and post improvements are evident, but perhaps some statistic should have been used to support it (i.e. t-test for related samples).

Idem in table 6.

DISCUSSION

The discussion is adequate.

Author Response

Point 1: Thank you for inviting me to review the paper on “Impact on physical fitness of the Chinese CHAMPS: a clustered randomized controlled trial”. This research is an example of the design of a good multi-component program for the promotion and improvement of health in schools. This is a good paper and deserves to be published in IJERPH.

The introduction clearly presents the research problem and the proposed objectives are adequate and subsequently well answered in the discussion.

The methodology is adequate and is supported by the previous publication of its protocol. The results provide the necessary data to respond appropriately to the objectives and use appropriate statistical methods.

The main criticism of this paper is that it is perhaps too ambitious in presenting results. There are many variables treated and difficult to follow for the reader.

 Response 1: Thank you for your positive comments and suggestions.

As for the comment on publishing the findings in multiple manuscripts, the physical fitness was the primary outcome and measured by one instrument. We believed that it would be more informative to present all components of the physical fitness outcome in one manuscript. We plan to publish the findings of secondary outcomes in the future.

Point 2: It is to consider positively the registry of the present investigation in Chinese Clinical Trial Registry. Could it also have been carried out in the International Clinical Trials Registry Platform?

Response 2: We only registered our trial in ChiCTR which is a recognized registry of the WHO Registry Network (https://www.who.int/ictrp/network/primary/en/), and meets the requirements of the ICMJE.

Point 3: Obviously before a study with such a multitude of variables it is necessary to provide many details even in the summary. Therefore, perhaps the summary is somewhat more extensive than desired.

 Response 3: Revised as suggested.

Point 4: Reformulate some keywords (i.e. Physical function training program; Multi-component promotion; School physical education policy). Some of them would not really help in the search processes in the databases. Try to make them clearer and more direct.

Response 4: Revised as suggested.

Point 5: Sometimes the acronym appears for the first time without detailing the corresponding word, i. e. line 66 RCT y line 74 CRF

Response 5: Revised as suggested.

Point 6: Obviously before a study with such a multitude of variables it is necessary to provide many details even in the summary. Therefore, perhaps the summary is somewhat more extensive than desired.

Response 6: We have made a concerted effort to reduce the presentation of the intervention protocol in this manuscript, since we have previously published the study protocol with detailed description of the rationale and design of the intervention in this journal. We would appreciate specific suggestions on which section(s) of intervention description should be reduced.

Point 7: The information and variables presented for a single article are too many. (Fitness, measure of levels of physical activity and sedentary behavior at the beginning and the end of the intervention, observation of physical education clases, food habits checklist, nutrition knowledge test, sedentary, rate monitors, behavior survey, demographic survey including their pubertal stage, development Scale, parents demographic information, etc).

It might have been easier to present it in two different publications.

Response 7: See our response to comment 1 from Reviewer 1.

Point 8: In table 2, put down the meaning of M and F. We assume that it is Males and Females but it would be appropriate to detail these issues at the bottom of the table.

Response 8: Revised as suggested.

Point 9: In Table 5, Student’s Nutrition Knowledge and Food Habits and Self-Reported Sedentary Behavior at Baseline and Posttest can be observed. Pre and post improvements are evident, but perhaps some statistic should have been used to support it (i.e. t-test for related samples).

Resonse 9: Added the statistics as suggested.

Point 10: Idem in table 6

Response 10: Added the statistics as suggested.

Point 11: The discussion is adequate.

Response 11: Appreciative of the positive comment.

Reviewer 2 Report

The article is written in English using clear and unambiguous text.

The article is not include sufficient introduction and background to demonstrate how the work fits into the broader field of knowledge. Relevant prior literature is not appropriately referenced.

Author Response

Point 1: The article is written in English using clear and unambiguous text. 

Response 1: Appreciative of the positive comment.

Point 2: The article is not include sufficient introduction and background to demonstrate how the work fits into the broader field of knowledge. Relevant prior literature is not appropriately referenced.

Response 2: Revised as suggested [see paragraph 1 of Introduction].

Reviewer 3 Report

The manuscript by Zhou et al., is a sound analytical manuscript exposing the effect of a multi-faceted intervention program for middle school students to reduce sedentary habits. The article is well conducted with the presentation of results in tables 4, 5 and 6. I have minor/major comments that the authors should address:

1) The main point is the narration of results based on the information provided in Tables 4 to 6. The description is quite poor and the section 3 (Results) should present a fully, well organised, presentation of results. Note that the actual poor description makes the abstract not to be very original since the abstract do no present the significant results of the manuscript. Therefore, the authors should aligned the description of results with the abstract. In contrast, the discussion section is well contextualized and the study strengths and limitations are well discussed.

2) Conclusions are very poor. This is due to the short presentation of results. Therefore, I recommend the authors to present a sound conclusion section based on the results. In addition, the conclusions should referred to the hypothesis of the manuscript.

3) The hypothesis of the manuscript are not well presented. They are embedded in section 3.1. but with no clear definition. Please, provide the 3 hypothesis early in the manuscript with clear sentences. Specify the hypothesis at the end of the Introduction section.

4) The authors use the term 'Chinese CHAMPS' several times. I recommend to remove 'chinese', since it does not add any scientific information.

5) Be aware of some mistakes in line 33 (remove the), 79 (should be CHAMPS), Table 2 (columns of all sites), Table 3 (ASP in first row), Table 3 (do not separate Postgraduate), Table 4 (do not separate Control).

6) I am wandering if Tables 5 and 6 should be only presenting percentages. Otherwise is quite difficult to follow the quantity of information they present.

Author Response

Point 1: The main point is the narration of results based on the information provided in Tables 4 to 6. The description is quite poor and the section 3 (Results) should present a fully, well organised, presentation of results. Note that the actual poor description makes the abstract not to be very original since the abstract do no present the significant results of the manuscript. Therefore, the authors should aligned the description of results with the abstract.. 

 Response 1: Revised the presentation of the results as suggested [see section 3.1].

Point 2: In contrast, the discussion section is well contextualized and the study strengths and limitations are well discussed.

Response 2: Appreciative of the positive comment.

Point 3: Conclusions are very poor. This is due to the short presentation of results. Therefore, I recommend the authors to present a sound conclusion section based on the results. In addition, the conclusions should referred to the hypothesis of the manuscript. 

Response 3: Revised as suggested.

Point 4: The hypothesis of the manuscript are not well presented. They are embedded in section 3.1. but with no clear definition. Please, provide the 3 hypothesis early in the manuscript with clear sentences. Specify the hypothesis at the end of the Introduction section.

Response 4: Revised as suggested.

Point 5: The authors use the term 'Chinese CHAMPS' several times. I recommend to remove 'chinese', since it does not add any scientific information.

Response 5: The acronym of CHAMPS has been used numerous times in previous publications. Adding “Chines” helps to distinguish the study from others.

Point 6: Be aware of some mistakes in line 33 (remove the),79 (should be CHAMPS), Table 2 (columns of all sites), Table 3 (ASP in first row), Table 3 (do not separate Postgraduate), Table 4 (do not separate Control). 

 Response 6: Revised as suggested.

Point 7: I am wandering if Tables 5 and 6 should be only presenting percentages. Otherwise is quite difficult to follow the quantity of information they present.

Response 7: Removed the number of participants in each cell of the ordinal variables in Table 5.

Round 2

Reviewer 2 Report

The article has improved